# The Impact of Large Language Models on Programming Education and Student Learning Outcomes

Gregor Jošt *, Viktor Taneski  and Sašo Karakatič 

Faculty of Electrical Engineering and Computer Science, University of Maribor, Koroška Cesta 46,
2000 Maribor, Slovenia; viktor.taneski@um.si (V.T.); saso.karakatic@um.si (S.K.)
* Correspondence: gregor.jost@um.si

**Abstract:** Recent advancements in Large Language Models (LLMs) like ChatGPT and Copilot have led to their integration into various educational domains, including software development education. Regular use of LLMs in the learning process is still not well-researched; thus, this paper intends to fill this gap. The paper explores the nuanced impact of informal LLM usage on undergraduate students' learning outcomes in software development education, focusing on React applications. We carefully designed an experiment involving thirty-two participants over ten weeks where we examined unrestricted but not specifically encouraged LLM use and their correlation with student performance. Our results reveal a significant negative correlation between increased LLM reliance for critical thinking-intensive tasks such as code generation and debugging and lower final grades. Furthermore, a downward trend in final grades is observed with increased average LLM use across all tasks. However, the correlation between the use of LLMs for seeking additional explanations and final grades was not as strong, indicating that LLMs may serve better as a supplementary learning tool. These findings highlight the importance of balancing LLM integration with the cultivation of independent problem-solving skills in programming education.

**Keywords:** large language models (LLMs); ChatGPT; Copilot; programming education; React; debugging



## 1. Introduction

In recent years, the integration of Large Language Models (LLMs) into various domains has revolutionized the landscape of artificial intelligence (AI) and machine learning (ML). Particularly in the field of natural language processing, modern, transformed-based LLMs such as OpenAI's ChatGPT and Microsoft's Copilot have demonstrated remarkable capabilities in understanding and generating human-like text. Beyond natural language understanding, LLMs have also found applications in programming education, offering students access to vast repositories of code snippets, explanations, and debugging assistance.

While the potential benefits of leveraging LLMs in programming education are evident, it is crucial to understand the nuanced impact of their usage on student learning outcomes. Previous studies [1,2] have begun to explore this relationship, highlighting both the advantages and potential pitfalls of integrating LLMs into educational settings. However, it is essential to note that many of these studies were conducted before the widespread availability of modern transformer-based attention mechanisms [3]. LLM services, including ChatGPT, Claude, Mistral, and Gemini, are open to the non-expert public. Consequently, there is a need for updated research that considers the specific functionalities and implications of these advanced language models.

This paper seeks to contribute to the expanding body of knowledge on LLMs in education by focusing on the impact of informal LLM usage on undergraduate students' learning outcomes in programming education. It aims to dissect the relationship between LLM usage patterns and student performance, providing fresh insights into the role of LLMs in programming education and guiding pedagogical strategies in this area. To this end, we introduce the following research questions and hypotheses that frame our research:

**RQ1.** *What is the overall impact of Large Language Model (LLM) usage on the final grades of undergraduate students in programming courses?*

**H1.** *A higher average usage of LLMs for studying is negatively correlated with the final grades of undergraduate programming students.*

**RQ2.** *How does the use of LLMs for generating code, seeking additional explanations, and debugging specifically impact the final grades of undergraduate students in programming courses?*

**H2a.** *The use of LLMs for generating code is negatively correlated with student final grades.*

**H2b.** *The use of LLMs for seeking additional explanations does not significantly impact student final grades.*

**H2c.** *The use of LLMs for debugging is negatively correlated with student final grades.*

These research questions and hypotheses steered our exploration into how ChatGPT and similar LLMs impact programming education. Our goal is to provide detailed insights to help shape teaching strategies that effectively integrate LLMs, ensuring they support and promote student learning and skill development in programming.

The demand for empirical research assessing the impact of these advanced LLMs on programming education is essential. Previous papers, reviewed in the next chapter, largely conducted before these high-capability models were widely accessible, offer limited insights into the effects of LLMs integrated with the latest AI advancements. Thus, the main contribution of this study lies in its empirical analysis of how different usage patterns of modern LLMs—ranging from code generation to seeking explanations and debugging—impact learning programming. This examination not only expands our understanding of the educational implications of LLMs but also provides targeted insights that can guide educators in designing instructional strategies that leverage these tools effectively while nurturing essential problem-solving skills. By backing-up our findings with robust statistical analysis and a controlled experimental setup, this research offers important findings in the integration of AI technologies in programming education.

## 2. Research Background

Recent years have witnessed a surge in studies exploring the integration of LLMs in educational settings. These investigations have delved into the potential benefits and challenges of utilizing LLMs, such as ChatGPT, across various domains of learning. In this subsection, we provide an overview of existing research to contextualize the current study's focus on the impact of informal LLM usage on programming education and student learning outcomes. As this paper explores the use of modern LLMs accessible to the public without specialized hardware or ML knowledge, the research background specifically examines the use of modern LLMs, excluding any reference to self-hosted pre-ChatGPT models.

In a 2023 study [4], the transformative potential of LLMs in higher education was examined, emphasizing opportunities and challenges. The authors concluded that while LLMs offer personalized learning and on-demand support through customized learning plans, reduced human interaction, bias, and ethical considerations were observed. To mitigate these challenges, universities should integrate such models as supplements rather than replacements for human interaction, establish ethical guidelines, involve students in model development, and provide faculty training and student support. Hence, universities must balance leveraging LLMs for enhanced education quality with addressing associated challenges to ensure a high standard of education delivery.

With a focus on ChatGPT [5], 50 articles published in 2023 were analyzed, focusing on the original version based on GPT-3.5. While recognizing the potential of ChatGPT to enhance teaching and learning, the study highlighted shortcomings in knowledge and

performance across subject domains and identified potential issues such as generating incorrect or fake information and facilitating student plagiarism. To address these challenges, immediate actions were recommended, including refining assessment tasks to incorporate multimedia resources, updating institutional policies, providing instructor training on identifying ChatGPT use, and educating students about its limitations and the importance of academic integrity. It was noted that leveraging ChatGPT in teaching and learning could involve creating course materials and assisting with active learning approaches, albeit with a need for accuracy verification. However, challenges related to accuracy, bias, and student plagiarism must be addressed through proactive measures such as incorporating digital-free assessment components and establishing anti-plagiarism guidelines. Additionally, instructor training and student education on ChatGPT's limitations and academic integrity policies were deemed essential for effective integration into education.

Grassini [6] acknowledged the increasing prevalence of AI within the educational domain as well. Despite debates and technological limitations, AI's presence in education is undeniable and promises substantial transformations in teaching and learning methodologies. Central to ongoing discussions is the concern over AI's potential misuse, especially in academic assignments, leading some to advocate for bans on AI tools like ChatGPT in educational settings. However, others argue for integrating AI technologies into educational structures, emphasizing the need to address student dependency and implement guidelines to mitigate risks. The advancement of AI technology, exemplified by the evolution of ChatGPT, poses challenges to safeguarding against potential misuse. Rethinking assessment strategies was deemed imperative once again. On the other hand, integrating AI applications into education does offer students valuable hands-on experience while preparing them for an AI-dominated future. Negotiating AI's swift transformations entails developing effective strategies and customized training modules for teachers and students to maximize the benefits of AI tools in education. Failure to equip students with AI skills may leave them at a competitive disadvantage in the job market, underscoring the need for an educational framework that both employs and scrutinizes AI tools for students' benefit.

Similarly, this was noted in another study [7], where the author stressed that the emergence of ChatGPT as a tool in education makes training for faculty and students necessary in order to maximize its utility. It is important that educators familiarize themselves with ChatGPT's functions, including evaluating accuracy and distinguishing between text and idea generation. It is recommended that educators encourage students to use ChatGPT, fostering equal opportunity for idea development and improving writing skills. As ChatGPT evolves, universities may integrate it with learning management systems, and specialized academic versions may be developed. ChatGPT is expected to enhance creativity and critical thinking skills by contrasting generated ideas with original human input. As education adapts to technological advancements, students will require skills such as the critical evaluation of information and effective presentation, which will be assessed through methods like presentations and defending work in collaboration with ChatGPT. The paper provides practical examples for utilizing ChatGPT in academic writing and suggests adopting its techniques for academic research and publication. Universities and educators are encouraged to adapt these suggestions to suit their specific needs and courses.

As stated in another systematic literature review that focuses on using ChatGPT in education [8], ChatGPT has the potential to enhance the teaching and learning process by offering personalized learning experiences, improving student motivation and engagement, facilitating collaboration, and providing quick access to information. However, challenges such as teacher training, ethical considerations, accuracy of responses, and data privacy need to be further addressed.

Similarly to our research, article [9] focuses on the role of debugging in software development and explores the potential of ChatGPT as a tool in this process. ChatGPT, primarily known for its proficiency in generating high-quality text and engaging in natural language conversations, possesses lesser-known capabilities that are equally remarkable. It can

identify errors in code by analyzing it against its training data. However, caution is advised in its use for debugging, as it should be part of a comprehensive software development strategy. While ChatGPT's ability to learn from past debugging sessions and offer natural language suggestions can enhance code quality, it has limitations in domain knowledge or context awareness. Although ChatGPT can automate aspects of debugging and bug fixing, human review and testing remain crucial. Developers should view ChatGPT as a tool to complement their skills rather than replace them, using its suggestions as a starting point for further consideration and testing. Integrating ChatGPT with other debugging techniques can enhance the efficiency and effectiveness of the software development process, ultimately leading to higher-quality software delivery.

Comparable statements about the use of LLMs as an aid in the process of debugging software solutions can also be made for LLM tools other than ChatGPT, like GitHub's Copilot. Authors in [10] investigate Copilot's capability in code generation, comparing its outputs with those crafted by humans. Their findings demonstrate that Copilot can produce accurate and efficient solutions for certain fundamental algorithmic problems. However, the quality of its generated code heavily relies on the clarity and specificity of the provided prompt by the developer. Moreover, the results of this study suggest that Copilot requires further refinement in comprehending natural language inputs to effectively act as a pair programmer. While Copilot may occasionally fail to meet all prompt criteria, the generated code can often be seamlessly integrated with minor adjustments to either the prompt or the code itself, as demonstrated by studies [11,12]. Although Copilot is the most advanced AI-driven code completion tool [13] that has a very high percentage of correctly generated programs [14], and it proposes solutions that surpass those offered by junior developers and are comparable to human-generated solutions in terms of correctness, efficiency, reproducibility, and debugging efforts, the discernment of flawed or suboptimal solutions still necessitates the expertise of a seasoned developer [10]. Consequently, while Copilot can significantly aid software projects when utilized by proficient developers as a collaborative coding tool, its effectiveness diminishes if employed by individuals lacking familiarity with problem contexts and proper coding techniques.

Other authors also conclude that if not used properly, Copilot may not reduce the task completion time or may not increase the success rate of solving programming tasks in a real-world setting [15]. However, Copilot is strongly preferred for integrating programming workflow since Copilot often provides a good starting point to approach the programming task [11,16].

To summarize, most existing research in the field tends to focus on the pros and cons of LLM integration in education or software development. On the other hand, the study represented in this paper stands out by conducting empirical investigations into how the use of LLMs directly impacts learning processes. By shifting the focus towards empirical research, we aim to provide valuable insights into the practical implications of incorporating LLMs into educational settings. The emphasis on empirical investigation fills a crucial gap in the current literature and contributes to a more comprehensive understanding of the relationship between LLM usage and learning outcomes.

## 3. Materials and Methods

Thirty-two second-year undergraduate students participated in our study designed to explore the impact of informal LLM usage on learning outcomes in programming education. The selection of second-year undergraduate students as participants in this study was carefully considered and aligned with the research objectives. They were chosen based on their foundational knowledge in web development using HTML, CSS, and vanilla JavaScript, acquired through coursework undertaken in the first year of their undergraduate studies. Despite possessing this foundational understanding, they had not been formally introduced to React, a prominent JavaScript library widely utilized for building dynamic user interfaces [17]. This made them ideal candidates for examining the efficacy of utilizing LLMs in facilitating the learning of new technology.

This experiment, conducted over a period of ten weeks, was organized into distinct phases, as depicted in Figure 1.

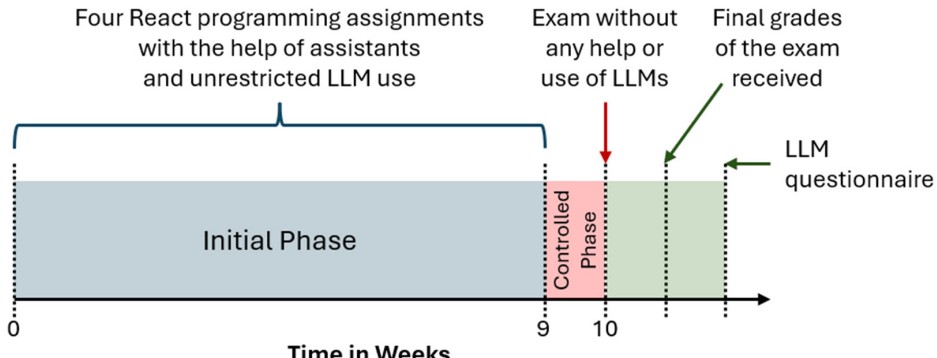

**Figure 1.** Overview of the whole experiment.

Given the increasing prevalence of LLMs in the programming domain, we provided students with the freedom to employ any LLM tool they deemed suitable for completing their programming assignments in the initial phase. By adopting this approach, we sought to mimic authentic programming environments where developers frequently leverage such tools to enhance their productivity and problem-solving capabilities.

To assess the influence of LLM utilization on students' learning experiences, we implemented a controlled phase where the usage of LLMs was prohibited. This phase was designed to isolate the effects of LLMs on learning outcomes by eliminating their presence during specific learning tasks. By comparing the performance and perceptions of students across the unrestricted and controlled phases, we aimed to observe the extent to which LLMs contribute to educational effectiveness.

### 3.1. Initial Phase

In the initial phase, spanning nine weeks, students were tasked with completing four assignments related to the development of React applications with Typescript. The assignments delved into various aspects of React development, covering topics such as component-based architecture, state management, and routing. Students were challenged to apply TypeScript's typing features to enhance code quality and maintainability throughout their assignments.

Each assignment presented unique challenges, progressively building upon the concepts introduced in the preceding tasks. So, the first assignment focused on building a basic React component hierarchy, while subsequent tasks explored more advanced topics like routing, state management, and lifting the state up, respectively. Furthermore, the assignments incorporated real-world scenarios to simulate industry-relevant experiences, encouraging students to develop problem-solving skills within the context of modern web development practices.

Throughout these assignments, participants were allowed to use LLMs informally for various purposes, such as bug identification, seeking additional explanations, or any other tasks that students deemed to be useful. Additionally, they received assistance from two experienced assistants (10+ years of programming experience) who provided support and guidance as needed. This informal usage, combined with the help from the assistants, mirrored real-world scenarios where students might employ AI-powered tools alongside human support as part of their learning process.

### 3.2. Controlled Phase

Following the completion of the four assignments spanning nine weeks, the study transitioned into its controlled phase in week ten. Participants were presented with a carefully crafted assignment where the use of LLMs was strictly prohibited. Conversely,

given the nature of the course (introduction to developing web applications using React), participants were allowed to use Google and official React documentation as supplementary resources to aid in solving the assignment. Within a two-hour time frame, participants were tasked with implementing an application using React. The assignment was intentionally designed to ensure that participants possessed all the necessary knowledge and skills to successfully complete the task. It mirrored the concepts and challenges they had previously encountered and mastered during the study. Importantly, the assignment did not introduce any new challenges or concepts; rather, it served as a practical application of their existing knowledge and skills. This approach aimed to create a fair and controlled environment for evaluating the impact of LLM's usage on participants' ability to independently implement a familiar task within a specified time frame.

Throughout the assignment, the two assistants closely monitored the process to ensure strict adherence to the no-LLM rule. Their presence helped maintain the integrity of the experiment by preventing any unauthorized usage of LLMs during the task.

Upon finishing the controlled assignment, participants were given a questionnaire to provide feedback on their study habits and implementation strategies throughout the experiment. This questionnaire included specific inquiries regarding their usage of LLMs, such as whether they utilized it for code generation, bug identification, or seeking additional explanations. Responses were measured on a five-point Likert scale to capture the extent and effectiveness of LLM usage in different aspects of their learning process. To mitigate any apprehensions about potential repercussions on their grades, it was ensured that students had already received their grades prior to the distribution of the questionnaire. This precaution aimed to alleviate concerns and encourage candid responses regarding their experiences with the LLMs.

Furthermore, participation in the questionnaire was voluntary; however, it was emphasized that non-participation would result in exclusion from the subsequent data analysis. This approach ensured a comprehensive dataset while respecting the autonomy of individual participants.

The questionnaire was designed in the Slovenian language to cater to the linguistic preferences of the participants. A translated version of the part of the questionnaire that is related to ChatGPT and its usage in the learning process is provided in Appendix A for reference. Additionally, Appendix B provides the instructions for the assignment used in the controlled phase.

To conclude, Appendix C breaks down the methods used in the experiment with the help of two detailed flowcharts. The first flowchart outlines procedures for the uncontrolled phase, while the second explains steps taken during the controlled phase of the experiment.

### 3.3. Alignment with Research Questions and Hypotheses

This methodology was thoroughly designed to address the research questions and hypotheses outlined in the introduction. By observing and analyzing the informal use of LLMs across various tasks and contrasting this with performance in a controlled, LLM-free environment, this study aims to explain the nuanced impact of LLM usage on undergraduate programming students' ability to independently solve programming tasks (RQ1 and RQ2). The structured approach of alternating between unrestricted and restricted use phases allows for a comprehensive examination of the potential benefits and drawbacks of LLM integration in programming education.

## 4. Results

### 4.1. Data Overview

To provide context for the results outlined in this section, we offer an overview of the code size associated with the tasks completed by participants during the experiment. The estimation of the approximate lines of code (LOC) for each task was derived from solutions prepared in alignment with the curriculum. The approximation of the LOC for each assignment is as follows: assignment 1 has 137 LOC, assignment 2 has 461 LOC,

assignment 3 has 427 LOC, assignment 4 has 921 LOC, and the final assignment (controlled phase) has 915 LOC. These values illustrate the approximate code size participants were required to write for each assignment, highlighting the variation in complexity and scope across the tasks.

Further on, an overview of the descriptive statistics of the measured variables used in the analysis is provided. Table 1 and the histograms in Figure 2 below offer a comprehensive overview of the extent to which the LLMs were utilized for various aspects of studying and the final grades given.

**Table 1.** Descriptive statistics of measurement.

|         |                        | Mean | Median | SD   | Min  | Max   |
|---------|------------------------|------|--------|------|------|-------|
|         | Generating code        | 2.59 | 2.50   | 1.10 | 1.00 | 5.00  |
| LLM use | Additional explanations | 3.75 | 4.00   | 1.24 | 1.00 | 5.00  |
|         | Debugging              | 3.78 | 4.00   | 1.16 | 1.00 | 5.00  |
|         | Average                | 3.38 | 3.67   | 0.94 | 1.00 | 5.00  |
|         | Final grade            | 6.72 | 8.00   | 3.10 | 0.50 | 10.00 |

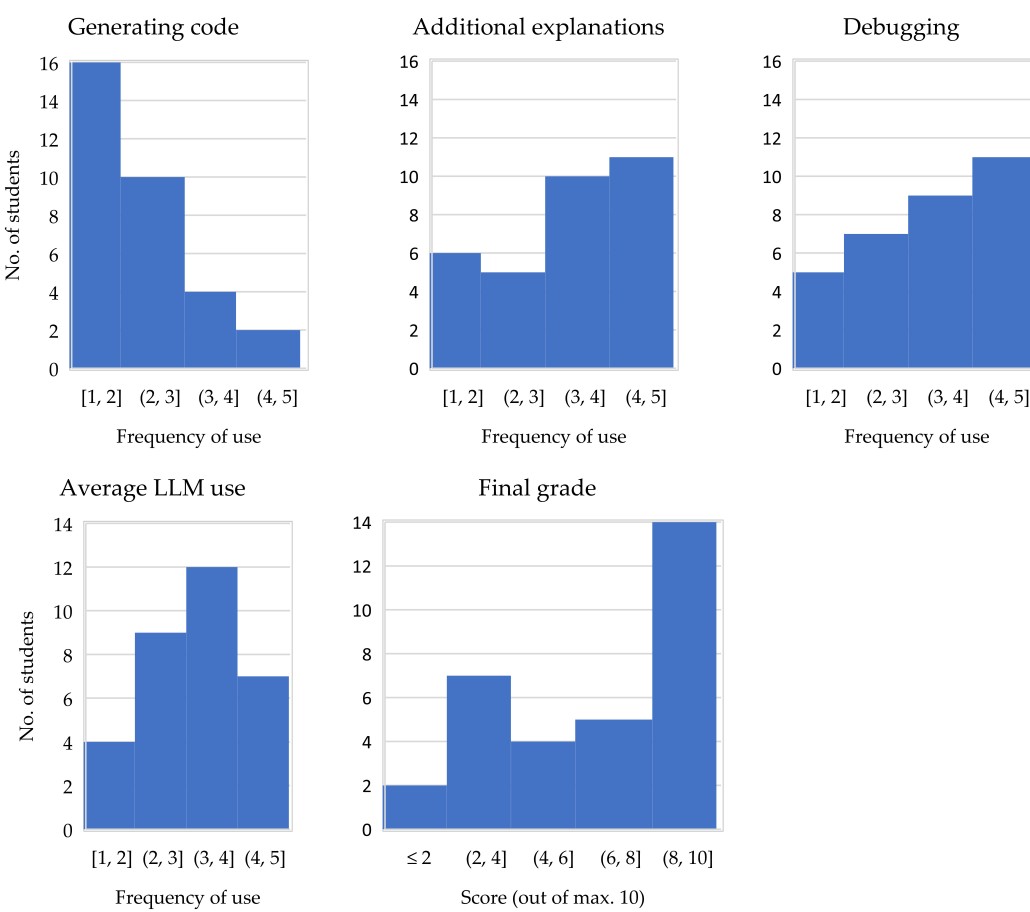

**Figure 2.** Histograms of the measurements about LLM use and the final grade.

Students reported a mean frequency of LLM usage for generating code at 2.59, with a median of 2.50. This is slightly above the lower midpoint of the scale, indicating that while some students did use LLMs to assist with generating code, it was not the most frequently utilized function. The standard deviation (SD) of 1.103 shows moderate variability in the use of LLMs for this purpose. The mean for seeking additional explanations is higher at 3.75, with a median of 4, suggesting that students were more inclined to use LLMs to gain further understanding of the material. The SD of 1.244 indicates a slightly higher variability

in response, with some students relying heavily on LLMs for explanations while others did not. The usage for debugging has a similar mean of 3.78 and median of 4, which indicates a comparable pattern of reliance on LLMs for this activity as for additional explanations. The SD of 1.157 denotes moderate variability among the students' responses. Overall, the average use of LLM across all activities has a mean of 3.38 and a median of 3.67. This points to a generally moderate use of LLM across the board, with the SD of 0.94 suggesting that the extent of LLM usage did not greatly differ among students.

The scatterplots provided in Figure 3 visualize the relationship between the final grades and their reported usage of the LLMs for different study activities.

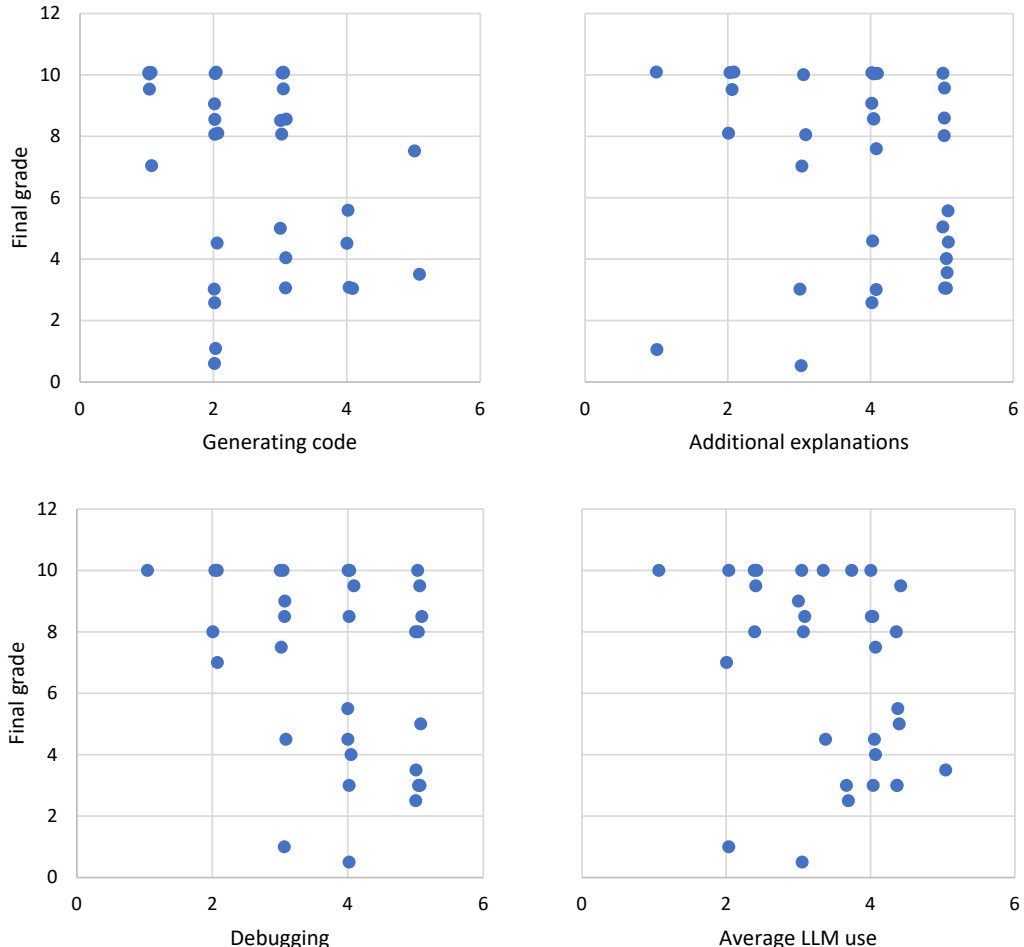

**Figure 3.** Scatterplots (with added jitter) of LLM use during learning process and the final grade.

There appears to be a spread of points that tends downward as the frequency of using LLM for generating code increases. This could suggest that students who used LLMs more frequently for code generation tended to receive lower final grades, in line with Spearman's rho value, which will be discussed in subsequent subchapters. For the use of LLMs for additional explanations, the distribution of points shows a less clear pattern, suggesting a weaker or potentially non-significant correlation between the use of LLM for additional explanations and the final grades, which corresponds with the non-significant *p*-value found in the Spearman's correlation analysis. Next, for the use of LLMs for debugging purposes, the scatterplot shows a trend where the higher usage of LLMs for debugging corresponds to lower final grades, which aligns with the significant negative correlation identified in the statistical analysis. Looking at the overall use of LLMs, there is a visible trend of decreasing final grades with increased average LLM use. This generalized trend encompasses all types of LLM usage and suggests that greater reliance on LLMs might be associated with poorer performance in the controlled assignment, supporting the idea

that while LLMs are helpful, they may also impede the development of independent problem-solving skills.

The observed correlations present in these scatterplots provide a visual representation of the potential impact of LLM usage on student performance. However, to determine whether these correlations are due to chance or reflect a genuine relationship, a comprehensive statistical analysis is required. The subsequent section will delve into this analysis, employing appropriate statistical tests to assess the significance and strength of these correlations, ensuring that the findings presented are robust and reflective of the true effects of LLM usage on learning outcomes.

### 4.2. Methodological Framework of the Statistical Analysis

The analysis of data collected from the experiment was conducted using R 4.3.3 in RStudio, focusing on the relationship between the final grade of the final assignment and the extent of LLM usage by students during their study process. Given the results of Shapiro–Wilk's test of normality, which indicated that all variables deviated from a normal distribution ($p < 0.05$ for all variables), a non-parametric method was deemed necessary for correlation analysis. Non-parametric methods, such as Spearman's correlation test, are preferable in these situations because they do not assume a normal distribution of the data. This approach is particularly useful when dealing with non-normally distributed data, as it allows for the identification of patterns and relationships without the need for the data to meet the assumptions of parametric tests. Spearman's correlation test, being a non-parametric method, is robust to non-normal data, making it suitable for our analysis.

Our research hypotheses were designed to explore the impact of LLM usage on academic performance, with H1 focusing on the general impact and H2a–c examining specific uses of LLMs. We hypothesized one-sided negative correlations based on the premise that while LLMs might offer immediate assistance, overreliance could potentially hinder the development of independent problem-solving skills. These skills are crucial for executing programming tasks without external aids. The choice of Spearman's correlation test aligns with our hypotheses by allowing us to test for these directional relationships. Spearman's rho, a measure of rank correlation, is particularly adept at identifying monotonic relationships between variables, which is essential for assessing the direction of the impact of LLM usage on academic performance. This methodological setup facilitated an in-depth examination of the directional impact of LLM usage on academic performance, particularly within the controlled environment of the final assignment where LLMs were explicitly prohibited.

### 4.3. Results and Interpretations

From a statistical standpoint, the analysis revealed varying degrees of correlation between LLM usage for different purposes (generating code, seeking additional explanations, and debugging) and the final grades of the participants. The Spearman's rho values indicated the strength and direction of these relationships, with negative values pointing towards an inverse relationship between LLM usage and final grades. The significance of these correlations was determined by the *p*-values, with values less than 0.05 considered statistically significant.

To further substantiate the robustness and reliability of these correlations, bootstrap confidence intervals were calculated for each correlation coefficient. Bootstrap analysis provides a non-parametric way to estimate the sample distribution of statistics based on random sampling with replacement. This method is particularly important in studies like ours, where the sample size is relatively modest. It allows us to estimate how Spearman's rho might vary due to sampling variability and, thus, provides a more comprehensive understanding of the reliability and stability of our results. The inclusion of bootstrap confidence intervals helps demonstrate the precision of the resulting estimates and highlights that the observed relationships are not artifacts of the random sample of students. Moreover, generating these intervals contributes significantly to validating the statistical

inference made in our study, presenting a clear picture of how certain we are about the existence and magnitude of the described correlations. The results of Spearman's correlation test and its bootstrap results are presented in Table 2.

**Table 2.** Results of Spearman's correlation analysis with bootstrap between LLM use and final grade.

| LLM Use | Spearman's Rho | 95% Bootstrap CI | $p$ |
|---|---|---|---|
| Generating code | −0.305 | (−0.595, −0.058) | 0.045 |
| Additional explanations | −0.201 | (−0.523, 0.220) | 0.135 |
| Debugging | −0.360 | (−0.628, −0.011) | 0.021 |
| Average | −0.347 | (−0.626, −0.044) | 0.026 |

Note: one-sided Spearman's correlation test was used, testing the negative correlations.

The results provided empirical support for H1, indicating a significant, though modest, inverse correlation between average LLM use and final grades (Spearman's rho = −0.347, $p$ = 0.026), suggesting that overall, higher engagement with LLMs may detract from the desired learning outcomes in programming education. This firm finding suggests that extensive engagement with LLMs might detract from the educational outcomes desired in programming education. The bootstrap confidence interval for this correlation (−0.626, −0.044) excludes zero, cementing the reliability of this result.

For H2a, the analysis revealed a significant inverse relationship between the use of LLMs for generating code and final grades (Spearman's rho = −0.305, $p$ = 0.045), supporting the concern that reliance on LLMs for code generation can undermine independent coding skills. The confidence interval (−0.595, −0.058) again excludes zero, confirming the reliability of these implications. Conversely, H2b's exploration of LLMs for seeking additional explanations did not yield a statistically significant impact on grades (Spearman's rho = −0.201, $p$ = 0.135), implying this form of LLM use might not hinder, and could potentially aid, student performance. The broad confidence interval (−0.523, 0.220) encompasses zero, reflecting this non-significance and corroborating the nuanced role LLMs play in educational settings. H2c was strongly supported by a significant inverse correlation between debugging with LLMs and final grades (Spearman's rho = −0.360, $p$ = 0.021), underscoring the importance of fostering independent debugging skills in programming education. The entirely negative confidence interval (−0.628, −0.011) robustly supports this outcome.

These results indicate a significant impact of LLM usage on learning outcomes in the context of programming education. The significant inverse correlation associated with code generation and debugging suggests that reliance on LLMs for these critical thinking-intensive activities could be detrimental to students' ability to independently solve programming tasks. This might imply that while LLMs can be a valuable resource for learning and problem-solving, their use needs to be balanced with the development of independent coding skills, especially in an educational setting where the ultimate goal is to foster self-sufficiency in problem-solving. On the other hand, the non-significant correlation for seeking additional explanations suggests that this type of LLM usage does not have a clear negative impact on student performance, potentially indicating that it serves more as a supplementary learning tool rather than a crutch that impedes skill development.

## 5. Discussion

The findings of this study shed light on the nuanced relationship between the use of LLMs in programming education and its impact on learning outcomes. Our analysis revealed distinct patterns regarding the frequency and way LLMs are utilized, particularly concerning code generation, seeking additional explanations, and debugging.

### 5.1. The Impact of LLMs on Code Generation and Debugging

Notably, our results demonstrate a concerning trend regarding the reliance on LLMs for code generation and debugging purposes. A significant negative relationship was found

between increased reliance on LLMs for tasks demanding critical thinking and decreased final grades, indicating a potential hindrance to students' ability to independently tackle programming challenges. This aligns with the assertion made by Haque and Li [9] regarding the importance of cultivating expertise in debugging to maintain the reliability and integrity of software systems.

Even though debugging and dealing with errors can be particularly difficult for programming novices and can often be a major source of frustration, it is still an essential skill in the context of programming since systematically examining programs for bugs, finding and fixing them is a core competence of professional developers [18]. Furthermore, the significance of debugging skills extends beyond the programming realm. Such skills are also common in our daily lives, and studies suggest that instructing debugging techniques can facilitate the transfer of these skills to non-programming contexts [19].

Our findings suggest that excessive reliance on LLMs for these tasks may hinder the development of essential troubleshooting skills, which are fundamental in software development. This is aligned with Pudari and Ernst [20], who stated that building software systems entails more than mere development and coding; it requires complex design and engineering efforts. While LLMs have made efforts to support coding syntax and error warnings, addressing abstract concerns like code smells, language idioms, and design principles remains challenging.

Moreover, the observed downward trend in final grades with increased average LLM use across all types of LLM usage further emphasizes the need for a balanced approach to integrating LLMs into programming education. While LLMs undoubtedly offer valuable support and facilitate learning, our results suggest that it is important to balance their use with the development of independent problem-solving skills. This resonates with the broader educational objective of fostering self-sufficiency in students, particularly in domains like programming, where autonomous problem-solving is crucial since it facilitates the process of learning to program [18].

### 5.2. Supplementary Learning through LLMs

Interestingly, our analysis also reveals a less pronounced correlation between the use of LLMs for seeking additional explanations and final grades. Unlike code generation and debugging, this type of LLM usage does not exhibit a significant negative impact on student performance. This suggests that leveraging LLMs for supplementary learning purposes may not impede skill development to the same extent as reliance on LLMs for critical thinking-intensive tasks. However, caution is warranted in interpreting these findings, as further research is needed to explain the nuanced role of LLMs in facilitating learning in programming education.

### 5.3. Educational Implications and Future Directions

The impact of LLM usage on programming education outcomes emphasizes the need for educators to carefully consider how these tools are integrated into learning experiences. Our study supports the implementation of LLMs as supplementary aids that can enhance understanding and engagement without undermining the development of critical thinking and problem-solving skills essential for programming.

However, it is crucial to approach the interpretation of these findings with caution. The less pronounced impact of LLMs on seeking additional explanations invites further investigation into how such tools can be optimally leveraged to support learning without compromising the cultivation of independent skills. Future research should explore the mechanisms through which LLMs influence learning processes and outcomes, identifying strategies that maximize their benefits while minimizing potential drawbacks. Another notable aspect that warrants further investigation is the correlation between students' previous grades and their utilization of LLMs. Exploring whether students who previously had lower grades tended to use LLMs more frequently could provide valuable insights into the dynamics between academic performance and LLM adoption.

In conclusion, this study underscores the importance of a balanced, thoughtful approach to incorporating LLMs into programming education. By carefully calibrating the use of these tools, educators can harness their potential to support student learning while ensuring the development of essential programming competencies. Continued exploration into the optimal integration of LLMs in educational settings remains a vital area of research.

### 5.4. Addressing the Balance between Productivity and Learning with LLMs

As discussed in Section 2 (Research Background), it is essential to acknowledge that LLMs can significantly enhance the productivity of software engineers, particularly in professional environments where routine tasks are streamlined. However, our study focuses on the implications of these tools in educational settings, where the primary aim is to build foundational programming skills.

Our findings suggest that while LLMs can improve efficiency in specific tasks such as syntax checking and understanding complex algorithms, their premature use can diminish essential problem-solving experiences. This can potentially hinder the development of the deep programming knowledge necessary for professional growth. Therefore, we advocate for a balanced approach to integrating LLMs into educational curriculums. By introducing these tools at later stages of programming education, after students have acquired basic coding principles, we ensure they benefit from both the productivity enhancements of LLMs and the critical problem-solving skills developed through traditional learning methods.

While LLMs are invaluable in increasing productivity in the professional setting, it is crucial to ensure their integration into educational programs does not compromise the development of fundamental programming competencies. This strategy prepares students to use these tools effectively in their future careers by having a robust skillset.

### 6. Conclusions

This study embarked on an exploration of the nuanced roles that informal usage of Large Language Models (LLMs) like ChatGPT plays in the learning outcomes of undergraduate students within programming education. Focused on a ten-week experimental framework involving thirty-two participants, this research specifically addressed how LLM usage impacts students' capacities for independent task implementation and knowledge acquisition in software development, with a particular emphasis on React applications. More specifically, our research shows the following.

RQ1: The analysis directly answered the primary question concerning the overall impact of LLM usage on programming education outcomes. We identified a significant negative correlation between the average use of LLMs and students' final grades (Spearman's rho = $-0.347$, $p = 0.026$), clearly suggesting that an increased general reliance on LLMs correlates with diminished academic performance in programming assignments.

RQ2: The study further dissected the impact of LLMs based on their specific uses—code generation, seeking additional explanations, and debugging. We found significant negative correlations for code generation (Spearman's rho = $-0.305$, $p = 0.045$) and debugging (Spearman's rho = $-0.360$, $p = 0.021$), supporting the hypotheses that these forms of LLM usage negatively affect students' ability to independently solve programming tasks. Conversely, the use of LLMs for seeking additional explanations did not significantly impact final grades (Spearman's rho = $-0.201$, $p = 0.135$), indicating its potential viability as a supplementary educational resource.

Our study highlights the need for the balance necessary in leveraging LLMs within programming education. While LLMs can undoubtedly serve as powerful tools for enhancing learning through supplementary explanations, their role in critical thinking-intensive tasks like code generation and debugging appears to negatively influence student outcomes. This separation underscores the imperative for educators to integrate LLMs into their pedagogical strategies thoughtfully, ensuring they augment rather than undermine the development of core programming skills.

While our findings provide valuable insights into the impact of LLMs on programming education, we acknowledge that our study is limited by the scale of the dataset. The relatively small sample size of thirty-two participants may not fully capture the broad spectrum of educational outcomes associated with LLM use in diverse educational settings. Additionally, the ten-week duration of our experimental framework may not adequately reflect long-term learning trajectories and the sustained impact of LLM usage on student capabilities. These limitations suggest the need for caution in generalizing our results across all programming education contexts. Further research involving larger sample sizes and extended study periods is essential to validate and refine our findings.

In the future, it is important that further investigations delve into the specific dynamics through which LLMs influence learning processes and outcomes. Such research should aim to refine strategies for integrating these technologies into educational frameworks, maximizing their benefits while mitigating potential drawbacks. Exploring the differential impacts of LLM usage across various learning styles and educational contexts will also be crucial for tailoring AI-enhanced learning experiences to diverse student needs.

Ultimately, a balanced approach that combines AI assistance with human guidance holds promise for optimizing learning experiences in the realm of software development and beyond.

**Author Contributions:** Conceptualization, G.J. and V.T.; methodology, G.J. and S.K.; validation, G.J. and S.K.; formal analysis, G.J. and S.K.; investigation, G.J., V.T. and S.K.; resources, G.J. and V.T.; data curation, G.J. and S.K.; writing—original draft preparation, G.J., V.T. and S.K.; writing—review and editing, G.J., V.T. and S.K.; visualization, S.K. All authors have read and agreed to the published version of the manuscript.

**Funding:** The authors acknowledge the financial support from the Slovenian Research and Innovation Agency (ARIS) (Research Core Funding No. P2-0057).

**Institutional Review Board Statement:** The study was conducted in accordance with the Declaration of Helsinki. Ethical review and approval were waived for this study due to the nature of the study, ethical review and approval were not required in accordance with the national and institutional guidelines.

**Informed Consent Statement:** Informed consent was obtained from all subjects involved in the study.

**Data Availability Statement:** The raw data supporting the conclusions of this article will be made available by the authors on request.

**Conflicts of Interest:** The authors declare no conflicts of interest.

## Appendix A. Questionnaire of LLM Usage during the Initial Phase

In this appendix, we provide a subset of questions extracted from the larger questionnaire specifically related to the implementation and learning process involving LLMs:

1. I used LLMs (ChatGPT, Copilot, Bing Chat, Gemini, or others) for code generation.
2. I used LLMs (ChatGPT, Copilot, Bing Chat, Gemini, or others) for providing additional explanation regarding coding challenges.
3. I used LLMs (ChatGPT, Copilot, Bing Chat, Gemini, or others) for debugging code.

These questions were essential for comprehensively understanding the diverse applications of LLMs within our study, illustrating their versatility in supporting various aspects of our research efforts. It should be noted that these questions constitute only a portion of a broader questionnaire designed to capture participants' experiences with LLMs throughout the implementation phase.

## Appendix B. Instructions for Assignment that Was Used in the Controlled Phase

The final assignment served as a comprehensive practical exercise that consolidated the skills acquired in the previous tasks during the initial phase of the study. The following

were the instructions used as a description for the assignment in the controlled phase, and its quality presented the final grade measurement:

Implement an application for tracking TV shows using React. The application should allow users to track the TV shows they have watched and add new shows to their list.

Requirements:

- The application must have a home page displaying a list of TV shows. Each item on the list should include the show's name and status (watched or not watched). Use conditional rendering to display the appropriate state.
- Clicking on an individual show in the list should open details below the list, including the show's name and description.
- The application must have a form for adding new TV shows to the watchlist. The form should have fields for the show's name and description, with the status defaulting to "false". Use state to manage form data.
- Use props and state lifting to pass data between components.

Tips:

- Divide the application into smaller components and use composition to assemble the user interface.
- Test your components as you build them to ensure they work as expected. Plan the architecture of the application before starting coding to avoid getting stuck later on.
- React-router is not required for implementation.
- The structure of interfaces is your choice.

**Appendix C**

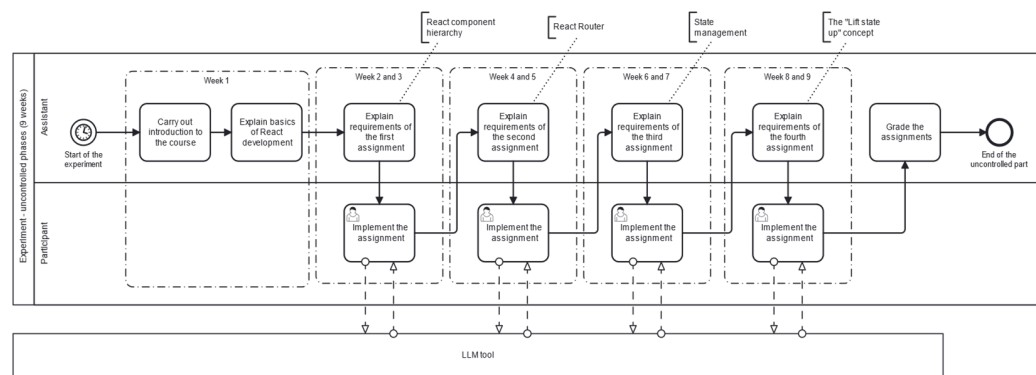

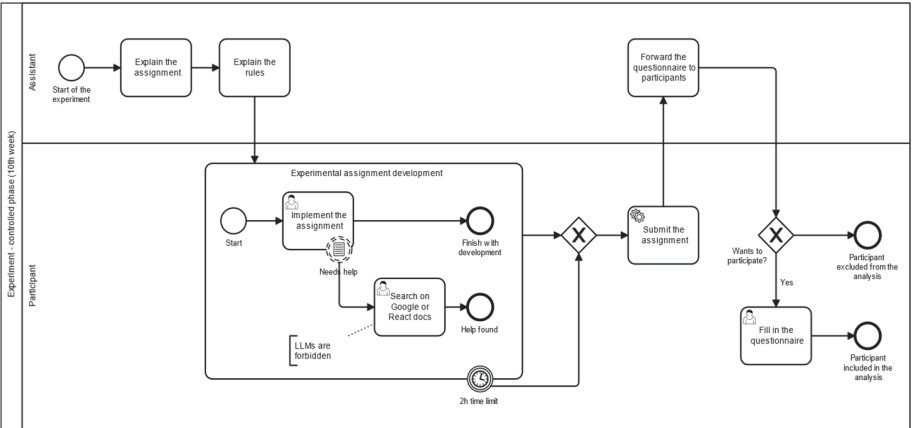

**Figure A1.** Flowcharts of the Method Used.

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
