# Peer review of "The Impact of Large Language Models on Programming Education and Student Learning Outcomes"

_applsci, doi:10.3390/app14104115_

Round 1

Reviewer 1 Report

Comments and Suggestions for Authors

The paper is devoted to the analysis of the impact of the use of the Large Language Models (LLMs) on undergraduate students' learning outcomes in software development education based on React applications.

Were the previous grades of the surveyed students analyzed? For example, the average grade from the previous semester?
Has it been studied whether students who previously had lower grades used the LLMs more often?
Please add appropriate discussion in the paper.

Please, discuss the limitations of the presented research in the conclusions.

Author Response

Comment 1: The paper is devoted to the analysis of the impact of the use of the Large Language Models (LLMs) on undergraduate students' learning outcomes in software development education based on React applications.

Response: Thank you for your comment regarding the analysis of students' previous academic performance.

Comment 2: Were the previous grades of the surveyed students analyzed? For example, the average grade from the previous semester? Has it been studied whether students who previously had lower grades used the LLMs more often? Please add appropriate discussion in the paper. Please, discuss the limitations of the presented research in the conclusions.

Response: In our current study, we did not incorporate students' past grades due to the absence of their explicit consent for accessing such information. However, your recommendation to include self-reported data on previous grades in subsequent research endeavors is insightful and warrants consideration, which we also added to the conclusion of the paper. We acknowledge the potential value this could add to our understanding of the usage of LLMs among students with varying academic backgrounds. We appreciate your constructive feedback.

 We extended the paragraph about future research (section 5.3) with the following:

 “Another notable aspect that warrants further investigation is the correlation between students' previous grades and their utilization of LLMs. Exploring whether students who previously had lower grades tended to use LLMs more frequently could provide valuable insights into the dynamics between academic performance and LLM adoption.”

Reviewer 2 Report

Comments and Suggestions for Authors

1. The English writing of this manuscript of the paper is still good, but the following two sentences may have grammatical problems that need to be corrected:

- Page 1, line 18: However, the correlation between LLM usage for seeking additional explanations and final grades was less pronounced
- Page 10, line 373: The significant negative correlation identified between increased usage of LLMs for these critical thinking-intensive activities and lower final grades underscores the potential detrimental effect on students' ability to independently solve programming tasks.

2. The conclusion of this manuscript shows that during the programming course, excessive reliance on LLM for code writing and debugging will lead to a decrease in students' final grades; however, students use LLM to understand program codes will not. This manuscript provides some good insights into how to properly apply LLM in programming courses.

3. It is known that LLM can increase the programming productivity by 50% for the programmers working in enterprises. Therefore, in the actual employment situation in the future, it should be the norm for programmers to use LLM to help write programs. However, this scenario is not the same as the method used in this paper to finally verify student learning results. Therefore, can the authors try to find some cases from your research data that can improve student productivity through LLM, but at the same time will not have negative effects on students' learning outcomes ?

4. Can the authors provide an approximate code size for each assignment during the experiment in this manuscript ? Approximately how many lines of code do students need to write for each assignment?

Author Response

Comment 1: The English writing of this manuscript of the paper is still good, but the following two sentences may have grammatical problems that need to be corrected:

- Page 1, line 18: However, the correlation between LLM usage for seeking additional explanations and final grades was less pronounced
- Page 10, line 373: The significant negative correlation identified between increased usage of LLMs for these critical thinking-intensive activities and lower final grades underscores the potential detrimental effect on students' ability to independently solve programming tasks.

Response: Thank you for the valuable observation. We reviewed the whole paper for any grammatical problems and corrected the ones we found. Specifically, the two sentences were revised to the following version:

 Abstract, the sentence was revised to:

 “However, the correlation between the use of LLMs for seeking additional explanations and final grades was not as strong, indicating that LLMs may serve better as a supplementary learning tool.”

 Section 5.1, the sentence was revised to:

 “A significant negative relationship was found between increased reliance on LLMs for tasks demanding critical thinking and decreased final grades, indicating a potential hindrance to students' ability to independently tackle programming challenges.”

Comment 2: The conclusion of this manuscript shows that during the programming course, excessive reliance on LLM for code writing and debugging will lead to a decrease in students' final grades; however, students use LLM to understand program codes will not. This manuscript provides some good insights into how to properly apply LLM in programming courses.

Response: Thank you for your comment highlighting the nuanced findings of our manuscript regarding LLM usage in programming courses. We appreciate your recognition of the insights provided on the effective application of LLMs in educational settings.

Comment 3: It is known that LLM can increase the programming productivity by 50% for the programmers working in enterprises. Therefore, in the actual employment situation in the future, it should be the norm for programmers to use LLM to help write programs. However, this scenario is not the same as the method used in this paper to finally verify student learning results. Therefore, can the authors try to find some cases from your research data that can improve student productivity through LLM, but at the same time will not have negative effects on students' learning outcomes?

Response: Thank you for your valuable comment regarding the impact of LLM on programming productivity, especially in professional settings. We acknowledge and agree that LLMs can significantly enhance the productivity of software engineers, as demonstrated by existing research. Some studies (also cited in our paper) suggests that the most substantial productivity improvements are observed among experienced or senior engineers rather than juniors. The core intention of our paper is to caution against premature reliance on LLMs during the learning phase. Our results show that the programming knowledge gathered is lesser in some regards when LLM is used in the process, which could prevent programmers from advancing in their knowledge, which potentially prevents the greater gains of using LLMs. Therefore, we advocate for a balanced approach to LLM usage, encouraging its integration into educational curriculums to enhance digital literacy yet emphasizing the importance of developing a solid programming foundation without overreliance on these tools. This strategy aims to prepare students for the professional world, where the effective use of LLMs becomes an invaluable asset without compromising their core programming skills. As we see this as a very important point, we added the following part to the discussion of our paper. Thus, we extended the discussion with the following section:

“5.4 Addressing the Balance Between Productivity and Learning with LLMs

As discussed in Section 2 (Research Background), it’s essential to acknowledge that LLMs can significantly enhance the productivity of software engineers, particularly in professional environments where routine tasks are streamlined. However, our study focuses on the implications of these tools in educational settings, where the primary aim is to build foundational programming skills.

Our findings suggest that while LLMs can improve efficiency in specific tasks such as syntax checking and understanding complex algorithms, their premature use can diminish essential problem-solving experiences. This can potentially hinder the development of the deep programming knowledge necessary for professional growth. Therefore, we advocate for a balanced approach to integrating LLMs into educational curriculums. By introducing these tools at later stages of programming education, after students have acquired basic coding principles, we ensure they benefit from both the productivity enhancements of LLMs and the critical problem-solving skills developed through traditional learning methods.

While LLMs are invaluable in increasing productivity in the professional setting, it is crucial to ensure their integration into educational programs does not compromise the development of fundamental programming competencies. This strategy prepares students to use these tools effectively in their future careers by having a robust skillset.”

Comment 4: Can the authors provide an approximate code size for each assignment during the experiment in this manuscript? Approximately how many lines of code do students need to write for each assignment?

Response: Thank you for the comment, we agree with your suggestion as it provides valuable input into understanding the complexity of the experiment. We expanded Section 4.1 with the following:

“To provide context for the results outlined in this section, we offer an overview of the code size associated with the tasks completed by participants during the experiment. The estimation of the approximate lines of code (LOC) for each task was derived from solutions prepared in alignment with the curriculum. The approximation of the LOC for each assignment is as follows: assignment 1 has 137 LOC, assignment 2 has 461 LOC, assignment 3 has 427 LOC, assignment 4 has 921 LOC, and the final assignment (controlled phase) has 915 LOC. These values illustrate the approximate code size participants were required to write for each assignment, highlighting the variation in complexity and scope across the tasks.”

Reviewer 3 Report

Comments and Suggestions for Authors

I think this paper is interesting but some suggestions:

1 section 3 needs more detailed results and why you should select this method is needed to be carefully explained, and the theoretical analysis maybe need better consideration

2  section 4.2 also needs some detailed discussions 

3 the dataset is not enough to derive the conclusion, I think more data or results need to support the conclusion

4  the future work can be discussed and the chatgpt can be discussed on its education impact.

Comments on the Quality of English Language

I think this paper is interesting but some suggestions:

1 section 3 needs more detailed results and why you should select this method is needed to be carefully explained, and the theoretical analysis maybe need better consideration

2  section 4.2 also needs some detailed discussions 

3 the dataset is not enough to derive the conclusion, I think more data or results need to support the conclusion

4  the future work can be discussed and the chatgpt can be discussed on its education impact.

Author Response

I think this paper is interesting but some suggestions:

Comment 1: section 3 needs more detailed results and why you should select this method is needed to be carefully explained, and the theoretical analysis maybe need better consideration

Response: Thank you for the comment, and we agree that the method needed the additional explanation. Thus, we added the following paragraph in section 3:

“Given the increasing prevalence of LLMs in the programming domain, we provided students with the freedom to employ any LLM tool they deemed suitable for completing their programming assignments in the initial phase. By adopting this approach, we sought to mimic authentic programming environments where developers frequently leverage such tools to enhance their productivity and problem-solving capabilities.

To assess the influence of LLM utilization on students' learning experiences, we implemented a controlled phase where the usage of LLMs was prohibited. This phase was designed to isolate the effects of LLMs on learning outcomes by eliminating its presence during specific learning tasks. By comparing the performance and perceptions of students across the unrestricted and controlled phases, we aimed to observe the extent to which LLMs contribute to educational effectiveness.”

Comment 2: section 4.2 also needs some detailed discussions 

Response: Thank you for your feedback. We've expanded our explanation to highlight the choice of the non-parametric Spearman's test over parametric tests. Additionally, we've detailed the rationale behind using one-sided tests, aligning with our hypotheses that explore the negative impact of excessive LLM usage on academic performance. The rewritten section 4.2 is the following:

 “The analysis of data collected from the experiment was conducted using R 4.3.3 in RStudio, focusing on the relationship between the final grade of the final assignment and the extent of LLM usage by students during their study process. Given the results of the Shapiro-Wilk's test of normality, which indicated that all variables deviated from a normal distribution (p < 0.05 for all variables), a non-parametric method was deemed necessary for correlation analysis. Non-parametric methods, such as Spearman's correlation test, are preferable in these situations because they do not assume a normal distribution of the data. This approach is particularly useful when dealing with non-normally distributed data, as it allows for the identification of patterns and relationships without the need for the data to meet the assumptions of parametric tests. Spearman's correlation test, being a non-parametric method, is robust to non-normal data, making it suitable for our analysis.

Our research hypotheses were designed to explore the impact of LLM usage on academic performance, with H1 focusing on the general impact and H2a-c examining specific uses of LLMs. We hypothesized one-sided negative correlations, based on the premise that while LLMs might offer immediate assistance, overreliance could potentially hinder the development of independent problem-solving skills. These skills are crucial for executing programming tasks without external aids. The choice of Spearman's correlation test aligns with our hypotheses by allowing us to test for these directional relationships. Spearman's rho, a measure of rank correlation, is particularly adept at identifying monotonic relationships between variables, which is essential for assessing the direction of the impact of LLM usage on academic performance. This methodological setup facilitated an in-depth examination of the directional impact of LLM usage on academic performance, particularly within the controlled environment of the final assignment where LLMs were explicitly prohibited.”

Comment 3: the dataset is not enough to derive the conclusion; I think more data or results need to support the conclusion

Response: Thank you for your comment, and we agree that the mentioned limitations of our researched are not obvious and thus should be explicitly outlined. We added the following part to the conclusion to address this:

 “While our findings provide valuable insights into the impact of LLMs on programming education, we acknowledge that our study is limited by the scale of the dataset. The relatively small sample size of thirty-two participants may not fully capture the broad spectrum of educational outcomes associated with LLM use in diverse educational settings. Additionally, the ten-week duration of our experimental framework may not adequately reflect long-term learning trajectories and the sustained impact of LLM usage on student capabilities. These limitations suggest the need for caution in generalizing our results across all programming education contexts. Further research involving larger sample sizes and extended study periods is essential to validate and refine our findings.”

Comment 4:  the future work can be discussed and the chatgpt can be discussed on its education impact.

Response: Thank you for your comment, and we agree that the discussion needed more impact on education. Thus, we added the following discussion on the impact on professional programming and education:

“5.4 Addressing the Balance Between Productivity and Learning with LLMs

As discussed in Section 2 (Research Background), it’s essential to acknowledge that LLMs can significantly enhance the productivity of software engineers, particularly in professional environments where routine tasks are streamlined. However, our study focuses on the implications of these tools in educational settings, where the primary aim is to build foundational programming skills.

Our findings suggest that while LLMs can improve efficiency in specific tasks such as syntax checking and understanding complex algorithms, their premature use can di-minish essential problem-solving experiences. This can potentially hinder the development of the deep programming knowledge necessary for professional growth. Therefore, we advocate for a balanced approach to integrating LLMs into educational curriculums. By introducing these tools at later stages of programming education, after students have acquired basic coding principles, we ensure they benefit from both the productivity enhancements of LLMs and the critical problem-solving skills developed through traditional learning methods.

While LLMs are invaluable in increasing productivity in the professional setting, it is crucial to ensure their integration into educational programs does not compromise the development of fundamental programming competencies. This strategy prepares students to use these tools effectively in their future careers by having a robust skillset.”

Also, we extended the future research with the following:

“Another notable aspect that warrants further investigation is the correlation between students' previous grades and their utilization of LLMs. Exploring whether students who previously had lower grades tended to use LLMs more frequently could provide valuable insights into the dynamics between academic performance and LLM adoption.”